# Rings with Heraldic Motives from the Wiener Neustadt Treasure: Imitations of Medieval Signet Rings?

**Nikolaus Hofer**

Department of Archaeology, Federal Monuments Authority Austria, 1010 Vienna, Austria;
nikolaus.hofer@bda.gv.at

**Abstract:** In 2007, a late medieval hoard was found in Wiener Neustadt (Lower Austria), which is extraordinary in many respects: the circumstances of its discovery were curious, its composition extremely inhomogeneous in terms of date and quality, and last but not least, the reconstructable circumstances of its concealment seem to have sprung almost from a Hollywood screenplay. It is due to a memorable coincidence that this treasure could be processed quickly within the framework of a scientific research project. The results were already presented in 2014 in a comprehensive monograph. One aspect among the numerous oddities of this hoard is the high number of finger rings it contains, which in turn form a very inhomogeneous category of finds. A group of rings with heraldic motifs on the ring plate is particularly striking and are the focus of this article.

**Keywords:** treasure; ring; signet ring; heraldic motif; archaeology; crafts; Middle Ages; Wiener Neustadt; Austria

## 1. General Characterisation of the Wiener Neustadt Treasure

The Wiener Neustadt treasure was found in 2007 by a private individual in a garden in today's municipal area of Wiener Neustadt (Lower Austria)[1] and was entrusted to the Federal Monuments Authority of Austria in 2010.[2] The hoard contains jewellery and clothing accessories as well as tableware and some spoons but no coins. In total, the hoard comprises 149 individual objects with a total weight of about 2200 grams (Figure 1).[3]

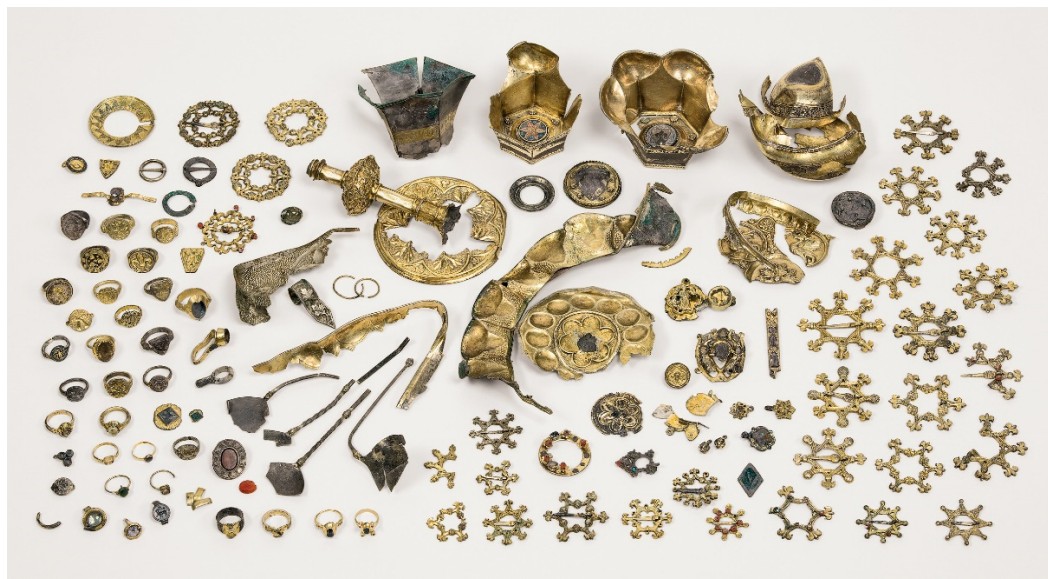

**Figure 1.** Treasure from Wiener Neustadt. Overall view. Photo: Paul Kolp; editing: Franz Siegmeth.

The 50 rings form the largest group within the treasure.[4] In addition to the rings with a ring plate, specimens with settings for gemstones and so-called fede rings (with a plastically executed hand motif) also appear. The next largest group within the overall complex are the brooches, which are characterised by a much stronger formal uniformity than the rings. In addition, individual hook fasteners, buttons, and belt fittings are also worthy of mention. Particularly striking are the fragments of high-quality dining utensils, some of which have inscriptions. Although older pieces are also present in all object categories, the majority of the preserved pieces can be dated to the 14th century, so that the treasure was probably concealed in the late 14th century or around 1400.

The material analyses have shown that almost all objects have silver as their main component and were originally gilded. Traces of use and, in many cases, evidence of manipulation prior to concealment were found on almost all the finds. Together with the temporal, formal, and also qualitative heterogeneity of the entire complex, this was interpreted by the processing team as a clear indication of a stock of 'old metal' that was probably intended for melting down or further processing (as 'raw material').

We can only speculate about the reasons for the hoard's concealment.[5] However, it is striking that the treasure was deposited at some distance from the medieval city walls of Wiener Neustadt, which was quite an important town in the Middle Ages and in the vicinity of the intersection of two important medieval roads (one of which—the road to Vienna—was, after all, the main route to Venice!). Owing to the consistence of the hoard and the presumable hiding place near an important trading route, a merchant and/or goldsmith is most likely to be the person who hid the treasure. The immediate proximity of the site to the location of the medieval gallows could indicate that this landmark was deliberately sought out to better retrieve the treasure. Finally, there is, very importantly, an explicit representation of a coat of arms on a drinking cup, which can be associated with a Wiener Neustadt alderman family. At least some of the objects may therefore have come from Wiener Neustadt itself, which in turn raises the question of why they were hidden outside the city. In any case, no major events that could have led to the hiding of the treasure—such as battles, epidemics, or pogroms—are known for the assumed time of concealment. Rather, the objects seem to have been deposited for personal reasons that can no longer be determined today.

## 2. The Rings with Heraldic Motifs

Among the 50 rings from the Wiener Neustadt treasure, 23 specimens—that is, almost half—have ring plates, with massive differences in the quality of execution and decoration.[6] There are simply cut silver rings next to elaborately executed examples with extensive gilding and the motifs are by no means homogeneous: in addition to heraldic subjects, which are discussed here in more detail, there are also pieces with a clearly 'personal reference' (hand gesture) or purely floral decoration.

The following decorative variants can be considered as heraldic motifs in the broadest sense: eagle, striding bird, rising bird, cock's head, lion, Agnus Dei, mythical creature, lily, blossom, crescent moon with a cross, human heads, and crossed hands. Almost all of the pieces can only roughly be attributed to the 13th/14th century on the basis of archaeological comparisons; a few pieces can be more precisely dated on the basis of palaeographic or art-historical criteria.[7]

The eagle motif (Figures 2–5) is most often represented in the finds (no less than four times). Two pieces show the heraldically 'regular' version facing right, whereas the other two have the eagle's head facing left (in the seal impression, this variant would then of course be heraldically 'correct').[8] The eagle has been one of the most popular heraldic motifs since Roman antiquity and is associated with numerous European ruling houses. It is noticeable that the ring plate of the ring (Figure 5) is shaped like a coat of arms. The ring plate with the motif of the rising bird (Figure 6), which is probably related to that of the striding bird (Figures 7 and 8), is also almost coat-of-arms-shaped. The latter is found on

two rather simple rings (each in combination with a pseudo-inscription), which are very similar to each other and find comparisons above all in Eastern and South-Eastern Europe.[9]

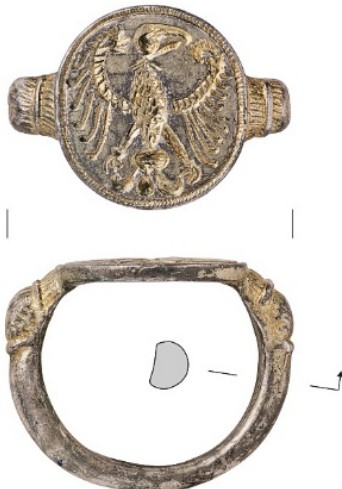

**Figure 2.** Treasure from Wiener Neustadt. Finger ring with "eagle" motif, 13th/14th century. Photo: Paul Kolp; editing: Franz Siegmeth.

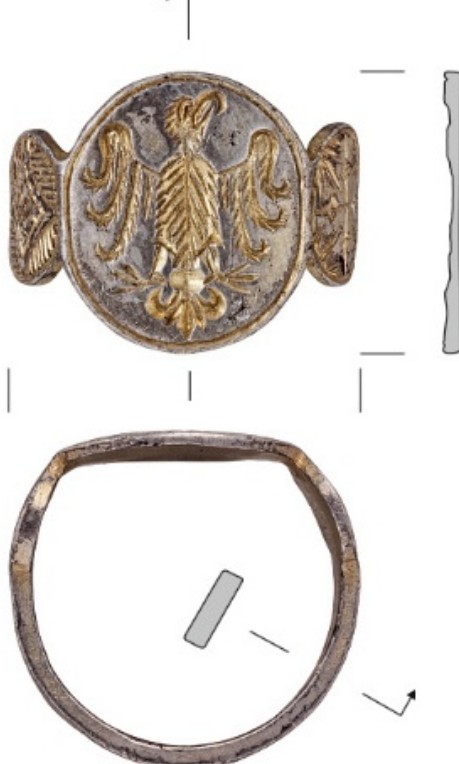

**Figure 3.** Treasure from Wiener Neustadt. Finger ring with "eagle" motif, 13th/14th century. Photo: Paul Kolp; editing: Franz Siegmeth.

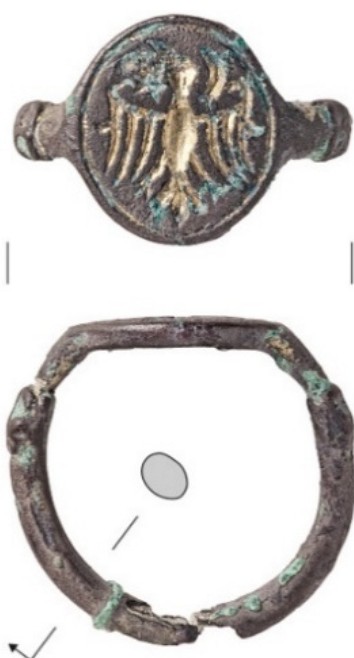

**Figure 4.** Treasure from Wiener Neustadt. Finger ring with "eagle" motif, 13th/14th century. Photo: Paul Kolp; editing: Franz Siegmeth.

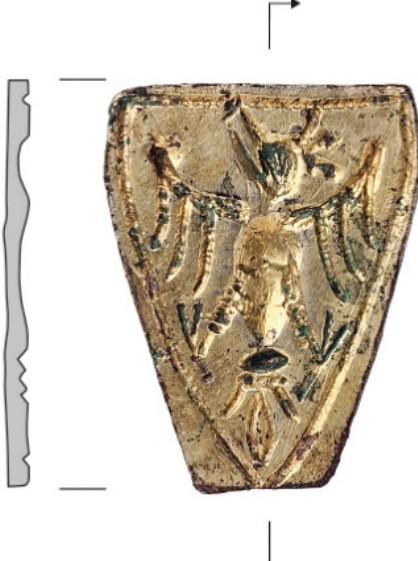

**Figure 5.** Treasure from Wiener Neustadt. Finger ring with "eagle" motif, early 14th century. Photo: Paul Kolp; editing: Franz Siegmeth.

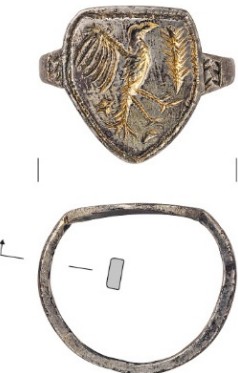

**Figure 6.** Treasure from Wiener Neustadt. Finger ring with "rising bird" motif, 13th/14th century. Photo: Paul Kolp; editing: Franz Siegmeth.

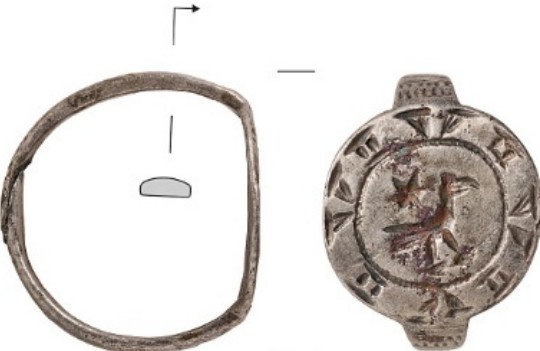

**Figure 7.** Treasure from Wiener Neustadt. Finger ring with "striding bird" motif, 13th/14th century. Photo: Paul Kolp; editing: Franz Siegmeth.

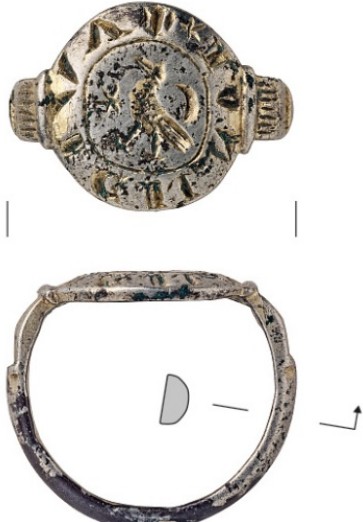

**Figure 8.** Treasure from Wiener Neustadt. Finger ring with "striding bird" motif, 13th/14th century. Photo: Paul Kolp; editing: Franz Siegmeth.

Other animal representations—also on shield-shaped ring plates—include a cock's head (possibly the head of a basilisk) (Figure 9) and a lion rampant (Figure 10), the latter alongside the eagle, one of the most popular heraldic animals in the Middle Ages.[10] Finally, there is a ring plate with a representation of the Agnus Dei (Figure 11)[11] and a ring with a mythical creature (perhaps just a 'failed' lion or wolf)[12] (Figure 12).

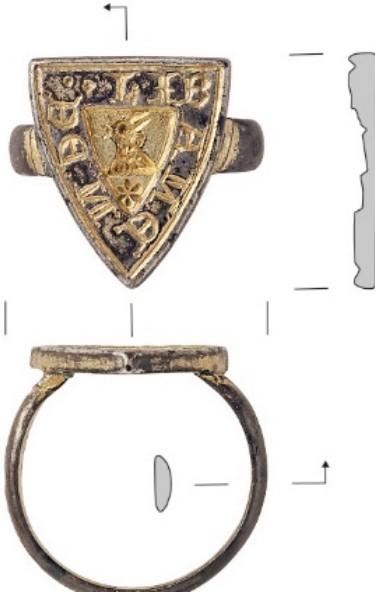

**Figure 9.** Treasure from Wiener Neustadt. Finger ring with "cock's head" motif, 13th/14th century. Photo: Paul Kolp; editing: Franz Siegmeth.

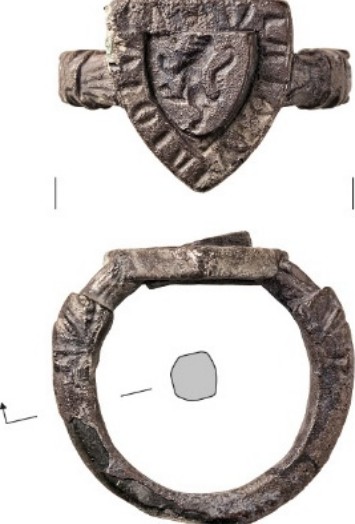

**Figure 10.** Treasure from Wiener Neustadt. Finger ring with "lion rampant" motif, 13th/14th century. Photo: Paul Kolp; editing: Franz Siegmeth.

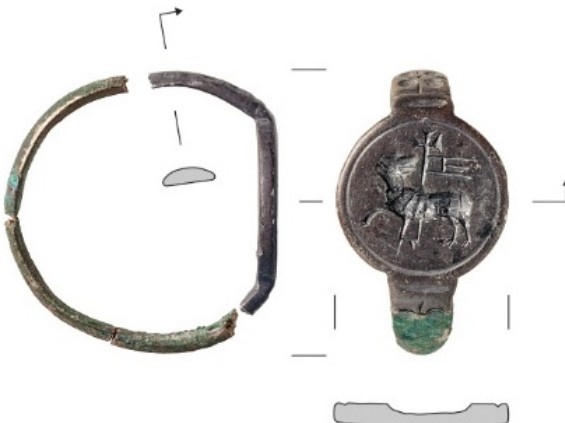

**Figure 11.** Treasure from Wiener Neustadt. Finger ring with "Agnus Dei" motif, 1st half of 14th century. Photo: Paul Kolp; editing: Franz Siegmeth.

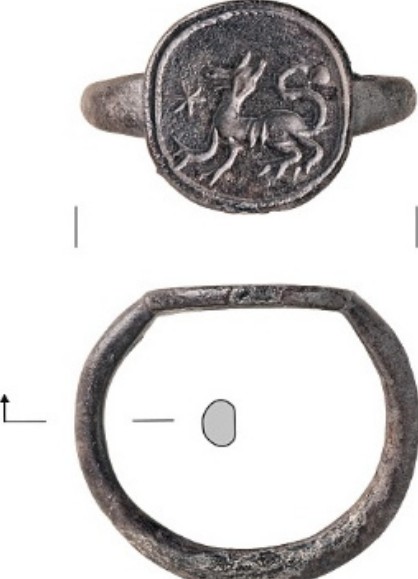

**Figure 12.** Treasure from Wiener Neustadt. Finger ring with "mythical creature" motif, 13th/14th century. Photo: Paul Kolp; editing: Franz Siegmeth.

Clearly heraldic references are also shown by the two rings with a lily motif (in both cases depicting the 'typically Florentine' filaments) (Figures 13 and 14) and the specimen with a blossom motif (Figure 15).[13] The ring with the moon-cross decoration (Figure 16)[14] is the only piece from the hoard similar to the seal of a citizen of Wiener Neustadt and could therefore refer to a specific heraldic device.[15] However, whether the two rings with representations of human heads (Figures 17 and 18) can also be regarded as heraldic motifs seems rather doubtful; in both cases, the decorative character, probably based on ancient coin images, is more likely to have been in the foreground.[16] Finally, Marianne Singer suggests that the clasped hands motif (Figure 19) could be interpreted as a representation of the coat of arms of the Franciscan Order[17], but in view of the handshake symbolism, which occurs several times in the hoard, an interpretation as a more personal 'gift of love' also seems possible.[18]

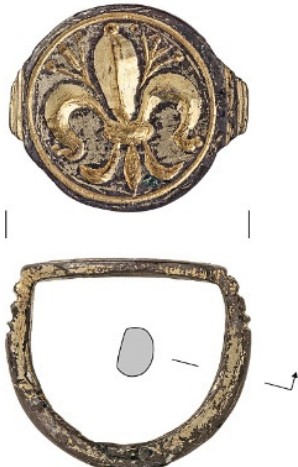

**Figure 13.** Treasure from Wiener Neustadt. Finger ring with "lily" motif, 13th/14th century. Photo: Paul Kolp; editing: Franz Siegmeth.

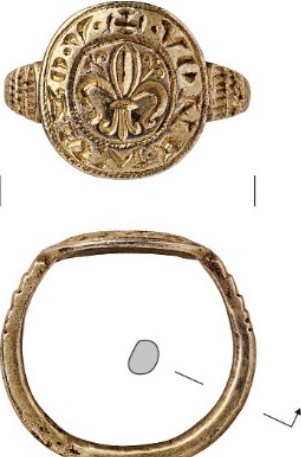

**Figure 14.** Treasure from Wiener Neustadt. Finger ring with "lily" motif, early 14th century. Photo: Paul Kolp; editing: Franz Siegmeth.

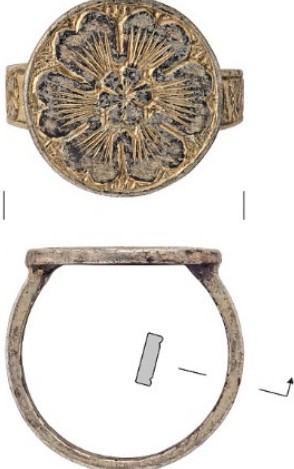

**Figure 15.** Treasure from Wiener Neustadt. Finger ring with "blossom" motif, 13th/14th century. Photo: Paul Kolp; editing: Franz Siegmeth.

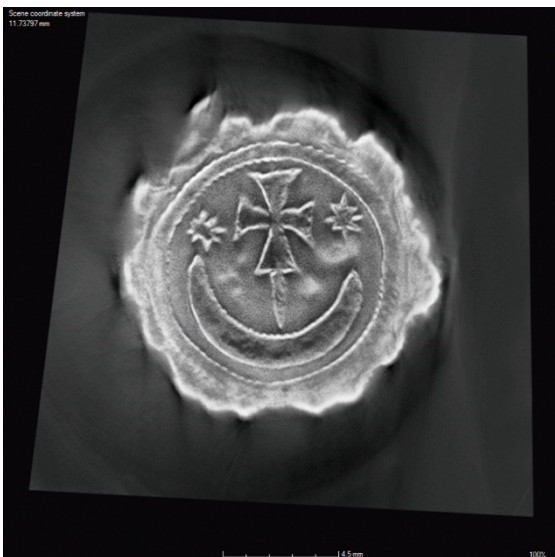

**Figure 16.** Treasure from Wiener Neustadt. Finger ring with "crescent moon and cross" motif, late 13th century. The motif is only visible here in the X-ray image, as the surface is heavily corroded. Photo: Paul Kolp; editing: Franz Siegmeth.

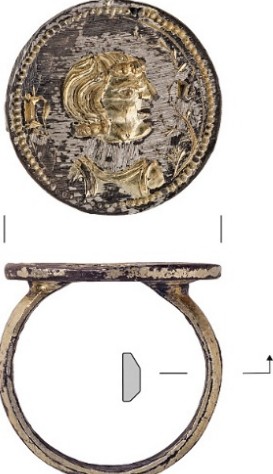

**Figure 17.** Treasure from Wiener Neustadt. Finger ring with "human head" motif, 1st half 14th century. Photo: Paul Kolp; editing: Franz Siegmeth.

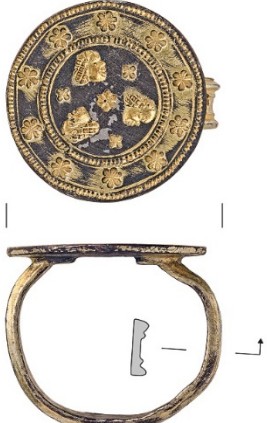

**Figure 18.** Treasure from Wiener Neustadt. Finger ring with "human heads" motif, 1st half 14th century. Photo: Paul Kolp; editing: Franz Siegmeth.

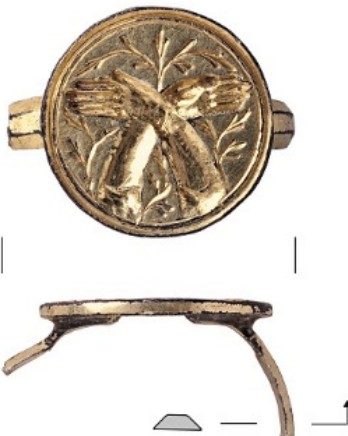

**Figure 19.** Treasure from Wiener Neustadt. Finger ring with "crossed hands" motif, 13th/14th century. Photo: Paul Kolp; editing: Franz Siegmeth.

### 3. Signet Rings or "Just" Prestige Objects?

The heraldic motifs on the rings from the Wiener Neustadt hoard led to the question of whether these rings might have been used for sealing documents. Andreas Zajic rightly points out that the subjects depicted do not represent real coats of arms but merely reflect a preference for heraldic symbols (which was common at the time).[19] Apart from the ring with a moon and cross motif already mentioned, which is, at best, a 'quotation' of a coat of arms with a pseudo-inscription, the only explicit representation of a coat of arms in the treasure is found on a medallion disc attached to the bottom of a cup. It was identified as the coat of arms of the Vierdung burgher family.[20] In seeking to explain the occurrence of heraldic subjects on finger rings, Andreas Zajic suggests that they were used for purely private "letter closure seals" (signets), presuming that the decorative character of the jewellery was not of primary importance for the owners anyway.[21]

The design of the objects provides strong arguments against their use as genuine seal or signet rings. The majority of the pieces in question were originally gilded on the surface and most of the representations are also very flat. In addition, most of the images would have been 'incorrectly' reproduced since the motifs on the seal impressions were not engraved back-to-front. Apart from that, the inscriptions—if present—are in most cases not legible and probably applied by the respective goldsmith without knowledge of their meaning.[22] In general, the inscriptions on the rings of the treasure are most commonly "mere sequences of letters without any recognisable meaning, and which cannot be understood as abbreviations [ . . . ] either".[23] And, last but not least, the ring plate on many specimens is so curved that it seems unsuitable for being used as a typar (seal stamp).[24]

Ultimately, this leads us to answer the question posed in the title of this article: whether or not the rings with heraldic motifs are imitations of 'real' signet rings. There is much to suggest that a clearly recognisable heraldic reference was important to the clients or buyers of these rings, even if it is rather unlikely that these rings were actually used for sealing—at least in the private sphere. It is possible that wealthy citizens of Wiener Neustadt, for example, wanted to demonstrate their closeness to the heraldic culture of the nobility (or simply their fashion sense)[25] by acquiring a ring with a coat-of-arms motif. This intention is perhaps clearest in the ring with the 'lion arms' (Figure 10), where the crest on the ring plate is even plastically emphasised by pivoting. However, it is also conceivable that these rings—as suggested by Andreas Zajic—could have been "objects of symbolic representation, [ . . . ] in that the depictions stood for the owner in some way".[26] This could be the case, for example, with the ring with the moon-cross decoration.[27]

Finally, in attempting to group the rings with heraldic motifs from the Wiener Neustadt treasure in terms of their possible use, only two (Figures 11 and 12) appear to be suitable as signet rings due to their largely flat ring plate, the lack of gilding, and the relatively

deeply incised motif; yet, they in particular do not show any distinct coat-of-arms motifs and also no circumscription.[28] In contrast, the rings shown in Figures 7–10, 14 and 16 are distinguished by the use of heraldic motifs in combination with (pseudo-)inscriptions, so that one could indeed think of imitations of signet rings in this case.

Since most of these rings were gilded on the surface, they are unlikely to have been used as typars but could have functioned as 'identification marks' of their owners. From the fact that many of the inscriptions on the rings are in fact 'pseudo-inscriptions' without any real meaning, one could also conclude that these pieces were primarily acquired by a group of buyers who were not able to read (or did not depend on it). In the third and largest group, which includes the rings shown in Figures 2–6, 13, 15 and 17–19, the heraldic motifs seem to have been primarily chosen for decorative reasons. The choice of subjects and the very high quality of the workmanship of some of these pieces clearly relate to the culture of the nobility.

However, this outlined classification remains purely speculative because the route of the individual pieces in the treasure of Wiener Neustadt, which, according to the current interpretation, is referred to as an 'old metal hoard', can no longer be reconstructed and thus the archaeological context is no longer an aid to interpretation. Even if rings of this kind are known as single finds, the large group of rings with heraldic motifs in the treasure of Wiener Neustadt represents a great peculiarity. Comparable find complexes with rings of this type are not known so far. The concrete significance of this special group thus remains to be clarified in future research and archaeology will perhaps make a significant contribution here, for example, through similar finds.

**Funding:** This research received no external funding.

**Data Availability Statement:** The study did not report any data.

**Conflicts of Interest:** The author declares no conflict of interest.

## Notes

1.   The treasure was discovered during the digging of a biotope on private property but was not recognised as such at first and it was stored in the cellar for years.
2.   Hofer (2014). This article is based on the results of the project.
3.   For the content of the treasure and the results of the interdisciplinary examination, see Kühtreiber et al. (2014).
4.   The treasure from Fuchsenhof (Upper Austria), for example, shows a similarly high number of rings: Krabath (2004, p. 233).
5.   See Hofer (2015) for this and the following in detail.
6.   See Singer (2014a, pp. 134–37) for the details.
7.   The dates were taken from the object catalogue: Singer (2014b).
8.   Singer (2014a, pp. 147–48) (with further literature).
9.   Singer (2014a, pp. 146–47) (with further literature).
10.   Singer (2014a, p. 149) (with further literature).
11.   Singer (2014a, pp. 149–50) (with further literature).
12.   For examples of similar motives see Singer (2014a, p. 148).
13.   Singer (2014a, pp. 150–51) (with further literature).
14.   Singer (2014a, pp. 155–56) (with further literature).
15.   Zajic (2014a, pp. 250–51) (with further literature).
16.   Singer (2014a, pp. 151–52) (with further literature).
17.   Singer (2014a, p. 152) (with further literature).
18.   For late roman examples cf. (von Bassermann-Jordan 1909, p. 37, Abb. 43; Johns 1996, p. 63, Figure 3.24, 3.25); and for medieval rings with this motif, for instance, Krabath (2004, pp. 276–78, 562–64, 588–89, no. 249–62, 272).
19.   Zajic (2014a, p. 244) (with further literature).
20.   For more details concerning the Vierdung family see Zajic (2014b, pp. 265–69).
21.   See Zajic (2014a, p. 247) with further discussion.
22.   For instance the inscription on the ring Figure 15: "WOLMICHEINRASGI". See Zajic (2014a, p. 248, fig 265).

[23] "Meaningless" inscriptions can be found on nine rings of the hoard: Zajic (2014a, p. 238).

[24] For examples of real signet rings see, e.g., Chadour and Joppien (1985, pp. 138–43, no. 213–22) (especially no. 222 with pseudo-inscription!) and Hindman (2015, pp. 180, 144, 212–13, 215, 218, no. 37, 38, 39, 43, 49).

[25] Heraldic motifs were very popular in the Gothic period and can also be found on other jewellery or costume items: Stürzebecher (2015, pp. 66–67).

[26] Zajic (2014a, p. 247) (with further literature).

[27] Maria Stürzebecher argues in the same vein regarding a ring from the treasure of Erfurt (Germany), which picks up a motif known from seals but it was certainly not used as a typar: Stürzebecher (2010, pp. 90–94).

[28] In the treasure from Fuchsenhof, too, only one (visually very similar) ring is explicitly addressed as a signet ring: Krabath (2004, p. 278; 625/Catnr. 317–18). A possible indication of the actual use of this object as a signet ring is the fact that the ring was deliberately broken in the middle (i.e., rendered useless or thus 'devalued' in the future). This is not the case with any of the finger rings from the Wiener Neustadt hoard.

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
