# Peer review of "Rings with Heraldic Motives from the Wiener Neustadt Treasure: Imitations of Medieval Signet Rings?"

_arts, 2022_

Round 1

Reviewer 1 Report

The article is written in a straight forward and clear manner, but it lives mostly from the illustrations which are indeed splendid. What is missing is a broader framework and a clear thesis. How does the study fit into studies of medieval jewelry and also this special ARTS issue. What is its truly new contribution compared to Hofer 2014 (esp. Zajic 2014) and Hofer 2015?

Edits/detailed remarks:

"treasure find" seems directly translated from German "Schatzfund"; replace by "treasure"

22    objects (instead of object)

41-47 give more details and eventually a map; and more references. It sounds as if this treasure was not professionally excavated - how do we know all the details about its deposition in relation to the medieval city (streets, gallows)? Also explain more why a goldsmith or merchant would have been the person to have held the treasure. The find spot near the gallows is not a convincing argument (could apply to anybody). The article implies the same further down, see lines 199-205; this seems inconsistent.

124-126 handshake motif as assign for a "gift of love" - give more comparanda; sine Roman times this is also a sign of fidelity/fides in non-romantic terms (e.g. ruler and military). Also, fig. 19 is not really a handshake-motif, but shows crossed hands in christian tradition with stigmata (see Hofer 2014, 152-153 fig. 151).

156  it was identified/it can be identified (instead of : it could be...)

160. "was not in the foreground" - do you mean: was not of primary importance?

170  "typar" - I am not sure this is the right term

190 inscriptions an pseudo-inscriptions: give a few examples to are your argument more specific

Reviewer 2 Report

The article is coherent, clear and well written. The main subject is clearly stated. All descriptions are informative. The great advantage of the article is numerous photos showing rings from the Wiener Neustadt Treasure, so the reader knows what he/she is reading about. The conclusions are convincing. The problem is that they are compelling, but at the same time, not enough embedded in historical and cultural background.

Let's have a look at the problem of pseudo-inscriptions. It is a popular subject amongst scholars interested in medieval culture. There are dozens of articles devoted to the functions of pseudo-Islamic, pseudo-Hebrew, and pseudo-Latin inscriptions commonly used as a motive in various paintings, sculptures, manuscripts, handcraft, etc. Primary literature is easy to find on the net (no need to search in library catalogues even). It takes no more than three minutes to find it: 1) http://www.garyschwartzarthistorian.nl/309-pseudo-semitism/; 2) https://www.cambridge.org/core/books/abs/hidden-language-of-graphic-signs/script-pseudoscript-and-pseudopseudoscript-in-the-work-of-filippo-lippi/3DDF979C21379C8798BE849125CC476F ; 3) https://grammarrabble.wordpress.com/2014/08/19/thoughts-on-psuedoscript-flora-ward/; 4) https://www.cambridge.org/core/books/viewing-inscriptions-in-the-late-antique-and-medieval-world/56CC02F057541EEAC046E1DF888B50F3; 5) https://www.ngv.vic.gov.au/essay/making-sense-of-nonsense-pseudo-script-on-an-italian-renaissance-maiolica-dish/. Really, it is possible to contextualize, even at the basic level, pseudo-inscriptions on rings from the Wiener Neustadt Treasure.

And now, the last phrase of the article: "The concrete significance of the rings with heraldic motifs thus remains to be clarified in future research - archaeology will perhaps make a significant contribution here, for example, through similar finds from burial or settlement contexts." It is highly improbable that rings from the Wiener Neustadt Treasure are exceptional. Once more: a simple search on the Internet convinces us that there are more similar medieval rings from all over Europe. One can even buy them at auctions. But the reader of the article has the impression that these rings are unique. Is it really so?

Reviewer 3 Report

I really enjoyed this article and am intrigued by the find. Thank you for including so high resolution images to support the text.

Author Response

Thank you for your comments.